# The Role of Diets and Dietitians for Para-Athletes: A Pilot Study Based on Interviews

**DOI:** 10.3390/nu14183720

**Published:** 2022-09-09

**Authors:** Hisayo Yokoyama, Miwako Deguchi, Nobuko Hongu

**Affiliations:** 1Research Center for Urban Health and Sports, Osaka Metropolitan University, 3-3-138 Sugimoto, Sumiyoshi-ku, Osaka-shi 558-8585, Osaka, Japan; 2Department of Environmental Physiology for Exercise, Graduate School of Medicine, Osaka Metropolitan University, 3-3-138 Sugimoto, Sumiyoshi-ku, Osaka-shi 558-8585, Osaka, Japan; 3Department of Nutrition, Graduate School of Human Life and Ecology, Osaka Metropolitan University, 3-3-138 Sugimoto, Sumiyoshi-ku, Osaka-shi 558-8585, Osaka, Japan

**Keywords:** interviews, survey, adult athletes, sports nutrition, dietary supports, dietary practices, wheelchair sports, dietitians

## Abstract

Efforts to provide nutrition support to para-athletes have not been established to date, and are far behind those established for athletes without disabilities. In the present study, we attempted to clarify the actual situation regarding dietary challenges of para-athletes. The aim of this study was to obtain clues to effective intervention methods that encourage the practice of sports nutrition. Six active elite para-athletes (30–70 years, four males) and a female physical therapist without physical disability participated in semi-structured interviews. All para-athletes had lower-limb disabilities and participated in the international wheelchair sports competitions (tennis, softball, and table tennis, with 2–26 years of player history). The interview items were on the ideal diet for improving competitive performance, evaluation of their typical diets, and the role of the dietitian as support. Responses obtained from participants were analyzed using quantitative content analysis by language analysis software. There are differences in the ideal diet based on the characteristics of the sport, but most participants believed that a nutritionally well-balanced diet with abundant vegetables was ideal for improving competitive performance. Para-athletes who use a wheelchair daily pay attention to their total calorie intake, because gaining weight is a critical issue for operating their wheelchairs and transferring themselves to and from their wheelchairs. Despite their world-class competition levels, none of them received routine dietary advice from dietitians. Some para-athletes did not even feel the need to engage with dietitians. Even for these para-athletes at a high level of competition, the “ideal diet” they considered was not always the optimal diet for improving their competitive performance. In addition, there are various barriers to practicing their optimal diet due to disability characteristics. Dietitians need to understand these barriers, their concerns and conflicts, and how to help them plan the optimal diet to improve their performance and maintain overall health.

## 1. Introduction

Japan’s third Sports Basic Plan, which covers the five-year period beginning April 2022, has just been formulated [1]. Under this plan, the Japan Sports Agency aims to build an inclusive, harmonious society for people with or without disabilities through sports. The plan sets a goal of increasing the weekly sports activity rate for people with disabilities to 40%, which was set as a goal of the second plan (April 2017–March 2022) [2]. Physically disabled people have low levels of non-exercise physical activity and a high prevalence of lifestyle-related diseases such as diabetes [3,4]. Thus, their acquisition of exercise habits for maintaining wellness is more demanding than people without disabilities. On the other hand, for people with disabilities, sports injuries sometimes impair residual physical function and threaten not only competitive performance, but also independence and quality of daily life [5]. Therefore, the need for medical support for para-athletes at various competition levels is expected to increase in the future.

Proper nutritional management based on sports nutrition improves the training outcome of athletes [6,7]. It helps metabolic adaptations to exercise such as acquiring and maintaining ideal body composition [8], enhancement of substrate oxidation ability [9], and contributing to early recovery from fatigue [10], conditioning, and healing sports injuries [11,12]. One of the most important concepts in sports nutrition is the “timing” of nutrition intake. It is necessary to strategically change intake depending on the “on-season” or “off-season”, or before or after the competition. It is critical for athletes to maintain their body weight at a weigh-in period. At international competitions such as the Olympic Games, it has become common for nutrition staff to accompany athletes without disabilities [13].

Traditionally, in Japan, sports for people with disabilities have been regarded as rehabilitation, while in other countries, sports for people with disabilities have been encouraged for their social participation [14,15]. However, in recent years, sports for people with disabilities have transitioned to competitive sports that pursue advanced sports performance, including the Paralympics [16]. Therefore, para-athletes need the benefit of sports nutrition, just as much as people without disabilities. Some dietitian associations such as Sports Dietitians Australia have issued position statements and guidelines on nutrition for para-athletes [17], and dieticians are working with the diverse athletes to expand their field of activity. However, efforts to provide nutrition support to para-athletes remain far behind athletes without disabilities. At present, there is very little evidence that can be referred to in supporting them from the nutritional aspect [18].

Broad laid out the key nutritional strategies for para-athletes in detail in her edited book [19]. She also mentioned in a review article the inadequate opportunities for nutritional support and nutritional education by practitioners for para-athletes [20]. Reports concerning the contribution of sports nutrition to the competitive performance, conditioning, and preventing injuries of para-athletes is scarce. Only 3.5% of certified sports nutritionists jointly certified by the Japan Dietetic Association and the Japan Sports Association profess para-sports and para-athletes as their field [21]. Para-athletes themselves lack knowledge of sports nutrition [20,22], and it is reported that many of them are unable to practice dietary intake to meet their energy and nutrient demands [23]. In addition, various dietary problems caused by the types of disabilities and individual differences of para-athletes make nutrition support challenging. Some of them have difficulty getting foodstuffs and in cooking their own meals, and others limit the amount of food and water they consume in order to avoid problems with excretion [24].

People with physical disabilities also may have problems with digestion and absorption due to primary diseases such as spinal cord injury and cerebral palsy. Thus, they have to frequently take small, divided meals [25]. Carbohydrate supplementation before, during, and after exercise is important for maintaining their competitive performance especially in people with spinal cord injuries at the cervical to upper thoracic spine level who can easily develop hypoglycemia during the game due to sympathetic nerve activity disorder [26]. Consumption of sugary drinks may bring them gastrointestinal symptoms such as vomiting and diarrhea [27,28]. Such digestion and absorption dysfunction may make it difficult to supply para-athletes with sufficient carbohydrates. Furthermore, though setting the energy requirements is the basis of nutritional management, determining the energy requirements is usually challenging for para-athletes [29]. For example, to estimate the number of calories a person needs to consume each day, it is necessary to track body weight changes to determine the total energy requirements. When someone is maintaining body weight, she/he is energy balanced. For gaining or losing their weight, she/he needs to adjust their total calories based on changes of their body weight. However, para-athletes need an individual customized measuring device for body weight measurement. Today, wearable devices with a 3D accelerometer which enables the user to estimate the energy expenditure during activities have become widespread. Unfortunately, this method is not easy for para-athletes. The device must be adapted correctly, such as when para-athletes are moving in a wheelchair [30].

Even if para-athletes and their coaches are aware of the need for nutrition support, very few sports organizations are providing this type of support for para-athletes [31]. It is not well understood how much sports injuries and poor conditioning are caused as a result of the lack of nutrition support for para-athletes [32]. Therefore, in order to practice sports nutrition for para-athletes, it is essential to develop teaching materials and guidelines that can be shared to para-athletes, instructors, and nutritionists, based on dietary problems attributed to the individual disability characteristics of para-athletes. We conducted a pilot study through interviews to identify the dietary issues that make it difficult for each para-athlete to practice sports nutrition, with the goal of contributing to the improvement of the competitive performance of para-athletes and preventing sports injuries through conducting proper sports nutrition management.

## 2. Materials and Methods

### 2.1. Participants

The survey participants were recruited and selected among the members of para-sports teams who regularly participate in various activities at the Osaka Prefectural Exchange Promotion Center for Persons with Disabilities, also known as Fine Plaza Osaka (Sakai City, Osaka, Japan, http://www.fineplaza.jp/ (accessed on 18 April 2022) [33]). Only members who met one of the following selection criteria were selected: (1) currently an active para-athlete with lower limb disabilities participating in international-level wheelchair events; or (2) a manager or trainer of a team to which he/she belongs. Athletes with only upper limb disabilities and those with hearing, visual, or intellectual disabilities were excluded. The staff from the Osaka Para Sports Association (Sakai City, Osaka, Japan, http://www.osad.jp/ (accessed on 18 April 2022) [34]) recruited candidates in person or by using a leaflet, and seven participants were selected for the survey.

The study protocol was approved by the Institutional Review Board of the Osaka City University Graduate School of Medicine (Approval no.: 2020-189). The nature and objectives of this study were explained to the participants orally and in writing, and written informed consent was obtained from all the participants. This study also conformed to the ethical guidelines of the 1975 Declaration of Helsinki.

### 2.2. Semi-Structured Interview Survey

Two interviewers surveyed each study participant based on an interview guide on the following three items (the following A, B, and C) during a 30–40 min semi-structured interview: (A) the ideal diet for improving competitive performance; (B) evaluation of their typical diet (given the ideal diet discussed in A); and (C) the role of the dietician as support. During the interviews, the interviewers were careful not to intentionally intervene when study participants were responding. At the end of the interviews, participants were also asked their age, gender, sporting event, sporting history, and primary disabilities.

All surveys were conducted in November 2020 during the COVID-19 pandemic. Therefore, the interviews were conducted in person with appropriate infection prevention measures, such as adequate social distancing and the use of masks by both the interviewer and the participant. The interviews were recorded on an IC recorder after obtaining permission from the survey participants.

### 2.3. Methods of Analysis

The content heard in the interviews was transcribed into text for each question item, and the text data analyzed using quantitative text analysis. First, a morphological analysis of the text data was conducted by breaking the sentences up into the smallest meaningful units, and then they were analyzed to classify the parts of speech. Second, a co-occurrence network schema was created to make the concepts of the data easier to understand. This schema connects the terms that are often used together in sentences and have similar occurrence patterns with arrows, which are then clustered together. The stronger the co-occurrence, the thicker the arrows, and the higher the frequency, the greater the circles representing them, thus helping to visualize the relationships between the terms in the data [35] (Figure 1, Figure 2 and Figure 3). The quantitative text analysis described above was performed using the language analysis software Text Mining Studio version 6.4 (NTT DATA Mathematical Systems Inc., Tokyo, Japan). Furthermore, the co-occurrence network schema was used to categorize the text data for each question item. The original sentences containing the terms that were comprised in a cluster were referred to, in order to identify the diet-related challenges of the survey participant [22,36].

## 3. Results

### 3.1. Overview of the Participants

The overview of the seven participants (i.e., six current active para-athletes and one physical therapist without disabilities) is listed in Table 1. The para-athletes are competing in wheelchair tennis (*n* = 1), wheelchair softball (*n* = 3), wheelchair table tennis (*n* = 2). Since the physical therapist works for a Boccia team, she is familiar with how players with problems with digestion and absorption due to cerebral palsy eat before and during games. We thought it would be useful for the study to interview her with such a perspective; therefore, we decided to include her even though she is not a para-athlete.

### 3.2. Basic Data and Frequency Analysis

The basic data regarding sentence composition of the text data in the responses to each question are presented in Table 2. Table 3 presents a list of the top 20 most frequently used terms that appeared in the responses to each interview question obtained from all respondents (Questions (A) and (C)) or from the six para-athletes (Question (B)). Following the direct expressions “consume” and “not consume” regarding eating behaviors, specific items such as “vegetables,” “side dishes,” “salad,” “protein,” and “information,” as well as terms related to the quality of the diet, such as “calories,” “nutrition,” and “balance” were frequently reported. Moreover, study participants expressed personal opinions about diet and dietitians, as “good,” “abundant,” “I don’t know,” “I don’t care,” and “want to learn”.

### 3.3. Co-Occurrence Network and Categorization of Response Details

#### 3.3.1. Interview Question (A): “What Is the Ideal Diet for Improving Competitive Performance?”

The co-occurrence network schema of the responses to the question (A) “What is the ideal diet for improving competitive performance?” is presented in Figure 1. The terms with similar occurrence patterns were classified into four clusters: (1) emphasis on vegetable intake and nutritional balance; (2) emphasis on comfort in daily life rather than focusing on the competitive performance; (3) different depending on the characteristics of the sporting event; and (4) I do not have a particular ideal diet; I do not care (Figure 1). Based on these clusters, the responses were categorized into four topics: (1) strategically consuming specific nutrients; (2) balanced intake; (3) meals that help in avoiding weight gain; and (4) absence or lack of knowledge on the ideal diet for improving competitive performance (Table 4). When the original sentences in the responses are referred to by category, there are differences in the ideal diet based on the characteristics of the sport, but all the study participants believed that a nutritionally well-balanced diet with abundant vegetable intake was ideal for improving competitive performance. However, some para-athletes did not seem to respond that way with clear understanding of how such a nutritionally well-balanced diet with abundant vegetable intake was helpful in improving their sporting event performance. Moreover, some para-athletes responded that their ideal diet aimed toward avoiding weight gain rather than improving their competitive performance. This was because they did not want weight gain to be an issue while operating wheelchairs or during movements, such as getting in or out of a wheelchair, i.e., their dietary choices were geared more toward ensuring functionality in daily living (Table 4).

#### 3.3.2. Interview Question (B): “How Do You Evaluate Your Typical Diet?”

The co-occurrence network schema of the responses to the Question (B) “How do you evaluate your typical diet?” is shown in Figure 2. The terms with similar occurrence patterns were classified into three clusters (Figure 2). Based on these clusters, the responses were categorized into three topics: (1) evaluation related to water intake; (2) specific evaluation comparing the ideal and actual eating behaviors; and (3) evaluation based on subjective perceptions (Table 5). When the original sentences in the responses are referred to by category, respondents seemed to be convinced that the necessary energy intake was lower among people using wheelchairs in their daily life than the able-bodied individuals. Many respondents thought that they were properly self-managed if they could reduce the amount of food they eat. Furthermore, the respondents believed that their diets were adequate based on subjective perceptions, such as “good physical condition” and “not getting tired easily.” However, a peculiar observation was that the majority of the study participants mentioned topics related to water intake, and the majority believed that their water intake level was insufficient (Table 5).

#### 3.3.3. Interview Question (C): “What Role Does the Dietician Play as Your Supporter?”

The co-occurrence network schema of the responses to the Question (C) “What role does the dietician play as your supporter?” is presented in Figure 3. The terms with similar occurrence patterns were classified into three clusters (Figure 3). Based on these clusters, the responses were categorized into three topics: (1) I have the desire to receive a consultation regarding diet if the opportunity arises; (2) I do not have any association; and (3) I do not have any expectations (Table 6). When the original sentences in the responses are referred to by category, none of the respondents received routine advice on diet from a dietician, and they explained why it was not easy to receive such support. A number of responses indicated a desire to learn about an ideal diet or to have their diet managed not only for improving competitive performance, but also for maintaining health, if the opportunity to consult with a dietician arises. Some responses suggested skepticism as to whether they would be able to put the dietary advice into practice even if they had the chance to receive it (Table 6).

## 4. Discussion

The purpose of this study was to clarify the actual situation regarding dietary behavioral problems of para-athletes with the aim of obtaining clues to effective intervention methods that encourage them in the practice of sports nutrition. Based on the results of the interview survey of this pilot study, it became clear that most of the respondents prioritized not causing inconvenience in their daily life and focusing on a diet that does not cause weight gain, rather than a diet for improving competitive performance. In addition, the self-assessment of their eating habits depended not only on the diet, but also on whether or not they recognized that their hydration (i.e., amounts and timing of water intake) was appropriate. Furthermore, it is interesting to note that even at such a high competition level, none of them received routine dietary advice from a dietitian as common practice, and some did not even feel the need to engage with a dietitian.

In the nutritional management of athletes, the intake of macronutrients is given the highest priority because it supplies the substrate (energy source) consumed for physical activity and meets the nutrient requirements for muscle protein synthesis caused by training [37,38]. When para-athletes put too much emphasis on nutritional *balance*, it becomes difficult for these athletes to take on sufficient macronutrients that meet the increase in energy required by training. For this reason, many athletes and their nutrition support staff are always careful not to run out of carbohydrates and proteins. On the other hand, the para-athletes in this survey thought that it would be ideal to eat a lot of vegetables and to eat balanced meals containing plenty of vegetables and a variety of nutrients in order to improve competitive performance. In some sports, such as wheelchair table tennis, respondents thought that agility, reaction to the ball, and technical skills are more important than muscular strength. They also answered that it is not necessary to take on protein diligently and to bulk up muscles since there is less movement during a game than in the same sport for people without disabilities. People with spinal cord injury (SCI) are generally at high risk of obesity and lifestyle-related diseases [3,4], which may be the reason why the respondents were strongly conscious of the need for a well-balanced diet with abundant vegetables. However, it has been reported that many para-athletes are not able to meet their energy requirements [39]. Undernutrition not only reduces athletic performance, but also affects the onset and exacerbation of pressure ulcers [40], a common medical problem of para-athletes using wheelchairs, thus para-athletes have to be more aware to prevent a deficiency of macronutrients than athletes without disabilities.

Regarding carbohydrates (staple food), many of our respondents thought that their intake should be restricted in order to prevent weight gain as much as possible. They considered a diet that does not make them gain weight as ideal. They evaluated their diet as properly managed if they could reduce the amount of food that they usually consume. Effective intake of carbohydrates before and after training and before and after a game is essential, especially for the strategy of endurance sports [41,42]. Nevertheless, according to a previous study we conducted, only 3% of para-athletes were able to answer correctly about the proportion of carbohydrates in the diet required for glycogen loading [22]. One of our respondents was afraid that gaining weight may inhibit the best fit to his custom-made racing wheelchair for a wheelchair marathon. All six para-athletes of this survey used a wheelchair at all times, even outside of competition, and most of them are careful not to gain weight to avoid inconvenience *in daily life*, i.e., gaining weight may cause some difficulties in operating a wheelchair or in transferring themselves to a toilet or a car. 

Depending on the degree of disability, such as cerebral palsy, it is difficult to have sufficient carbohydrates before their game due to dysphagia and gastrointestinal dysfunction [25], and the respondent without disability in this study who supports players of international boccia competitions was concerned about how to prevent performance deterioration during a game due to hypoglycemia. However, most of the para-athletes in this study seemed to refrain from carbohydrate intake without knowing the amount of carbohydrates they should take. People with physical disabilities generally have challenges in setting target intakes of nutrients such as carbohydrates and the total energy intake. In general, the energy balance can be easily grasped by measuring body weight and body composition [43], but some para-athletes cannot get frequent measurements because they need an individually customized measuring device, such as a wheelchair-accessible weighing scale, if they cannot stand on a weighing scale. There are a few studies reporting methods for estimating the muscle mass and resting metabolic rate for people with physical disabilities [44,45]. Methods for estimating the exercise-induced energy expenditure using a three-dimensional accelerometer that takes into account the movement in a wheelchair have also been verified [46]. However, none of them have been fully established. Under these circumstances, we have no choice but to set the target intake of nutrients by trial and error according to the individual disability characteristics of para-athletes using the currently available methods of estimating energy balance.

It should be noted that the respondents in this study always paid attention to their fluid intake and evaluated whether their diet was appropriate based on their adequate fluid intake. Urinary tract infections are more likely to occur in people with SCI due to dysuria and the use of urinary catheters [47], and many of them think that it is effective to drink sufficient water to prevent these infections. In reality, water intake and hydration status itself are not risk factors for urinary tract infections. Established evidence suggests regular urination is needed to minimize residual urine for prevention of urinary tract infections in people with SCI [48]. Nevertheless, due to the difficulty of urination management including finding and accessing a multipurpose toilet, transferring to the toilet from a wheelchair, and putting on and taking off their clothes, para-athletes may have the awareness to reduce the frequency of urination, and as a result, most of the para-athletes in this survey evaluated themselves as having insufficient water intake. Proper hydration management is important not only for maintaining competitive performance, but also for preventing heat stroke [49]. In people with SCI, autonomic dysfunction reduces the ability to regulate skin blood flow and sweating, which are necessary for thermoregulation, making them prone to heat stroke [50]. Therefore, if possible, it is desirable to evaluate water intake based on body weight change and have morning urine specific gravity testing. In addition, urine color is a practical index of hydration status. Utilizing a urine color chart has a potential to be an effective measure to help para-athletes optimize water intake [24].

At international competitions, it is common for nutrition staff to accompany athletes [13]. These athletes have the opportunity to receive nutritional guidance and advice from sports nutritionists or dietitians, even for their daily trainings. However, the nutrition support for para-athletes is still inadequate compared to athletes without disabilities. In the present study, none of the para-athletes received nutrition advice from a dietitian routinely, even at high levels of competition, and some athletes did not feel the need to engage with a dietitian. A previous study of female wheelchair basketball players in Turkey reported that only 18% of the athletes recognized nutritionists as a source of information about nutrition [51], and our result was compatible with the previous study. 

Some respondents of our study answered that they would like to consult with a dietitian if the opportunity became available. They mentioned specific contents regarding nutrition and dietary intake methods as the information they would like to obtain from the dietitian (e.g., “the amount of energy and protein required (for me)”; “the cooking methods that do not impair nutrition”; “how to utilize designated healthy foods and nutritional supplementary foods”). On the other hand, other respondents answered, “I don’t know where to find a consult for nutritional guidance” or “[To consult a dietitian] is difficult in terms of cost”. These responses suggested that nutritionists are not familiar to them, and that nutritional guidance and nutrition education by nutritionists are not recognized by para-athletes as support that they can routinely receive. Furthermore, there were many skeptical opinions as to whether the tips informed from dietitians can be put into practice in their daily routines and habits without difficulty. 

They also regarded nutritional guidance as an experience of passive learning from a dietitian. Needless to say, nutrition support for para-athletes must be an interactive process in which para-athletes and dietitians share the nutrition challenges of each para-athlete, and work together to reach their goals [29]. It is desirable that a nutrition support system is established by making close partnerships between para-athletes and nutritionists who can manage nutrition in consideration of health problems such as digestive dysfunction and pressure ulcers. The physical therapist, without disability in our study, gave an interesting view of the issue that could be considered “crossing the lines for nutrition support” for para-athletes. Currently, often exercise practitioners (e.g., trainer, coach, physical therapist, etc.) who have more opportunities to interact with para-athletes on a daily basis are providing nutrition education to para-athletes. However, nutrition education is not their specialty area, and the sports nutrition information they are giving to para-athletes may not be up to date or evidence-based. Some nutritionists/dietitians may be providing updated nutrition support/guidance to para-athletes, but they may not know the para-athletes’ daily routines and habits. The best way to handle this situation is for exercise practitioners to collaborate with, or refer para-athletes to, nutritionists/dietitians. In this way, acting as an intermediary between professionals/specialists, para-athletes, and nutritionists/dietitians could ensure the best nutrition advice based on their scope of practice [52].

This study has some limitations that should be mentioned. First, since this survey was a pilot survey targeting a very small number of para-athletes and a physical therapist, it may be difficult to generalize our results. In particular, the respondents were distributed over a wide range of ages with different types and degrees of severity of their impairment. All para-athletes used wheelchairs in their daily lives, and the sporting events were limited, preventing us from sub-grouping para-athletes. The method of nutrition support to be taken may be different between wheelchair users and those who have amputations/defects who can stand or walk while wearing prosthetic limbs outside of competition. Next, this study did not evaluate the actual dietary content of the participants. Therefore, there is a possibility that the result of their dietary self-assessments does not match whether or not the energy and nutrient intake originally required has been satisfied.

Taking into consideration the results of this pilot survey, nutrition support methods categorized by age, the type of parasports, and the characteristics of the disabilities are needed. Therefore, it will be important to expand the scope of the survey to include other types of parasports and other characteristics of disabilities such as amputations/defects, and to explore the nutritional challenges of para-athletes in each field. In addition, it is necessary to further investigate the relationship between their dietary intakes and the results of their dietary evaluations. In the next full study, we plan to develop and validate energy requirement estimate models and to create dietary teaching materials and guidelines in order to establish a feasible nutritional strategy for para-athletes. We position well to conduct the studies mentioned above as our next research plan.

This study had several notable strengths. This study was one of the first to explore the interactions between para-athletes and health practitioners, i.e., dietitians, physical therapists, exercise professionals, etc. The data provided in this study help to fill some important gaps in the literature, providing better understanding of the interactions with important practical implications from the perspective of both para-athletes and practitioners. We objectively evaluated and visualized the results of the interview survey and understood what kind of dietary policies para-athletes have and how they practice their dietary intake on a daily basis using the method of quantitative text analysis. We identified several challenges and difficulties that para-athletes face when practicing sports nutrition. Finally, the result of this study could share a new perspective of dietary support for para-athletes with many dietitians involved in sports nutrition and contribute to establishing a better partnership between para-athletes and dietitians.

## 5. Conclusions

Several factors which are indispensable for effective nutrition support for para-athletes are summarized in Figure 4. Understanding any of these factors, the interaction between para-athletes and practitioners is placed at the most prioritized position; if there is no interaction, there will be no success supporting para-athletes. Even for para-athletes at a high level of competition, the “ideal diet” they consider is not always the optimal diet for improving their competitive performance. Even if they understand a truly desirable diet, there are various barriers to implementing practice due to the characteristics of their disabilities. Thus, there remains much room for nutrition support for them, and dietitians need to understand their worries and conflicts and act to help them practice sports nutrition.

## Figures and Tables

**Figure 1 nutrients-14-03720-f001:**
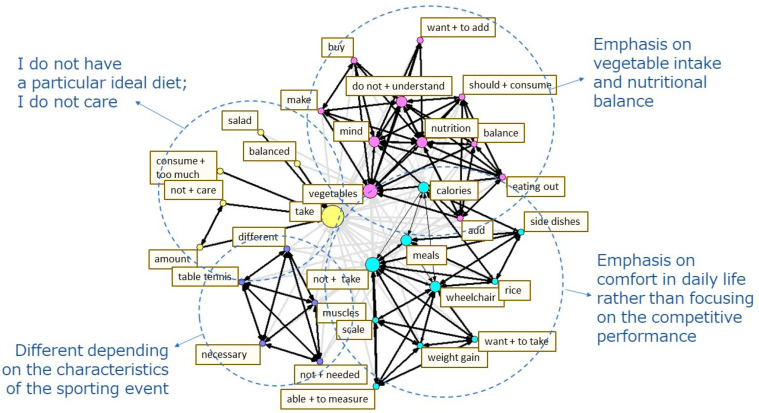
The co-occurrence network schema of the responses to the question “What is the ideal diet for improving competitive performance?”.

**Figure 2 nutrients-14-03720-f002:**
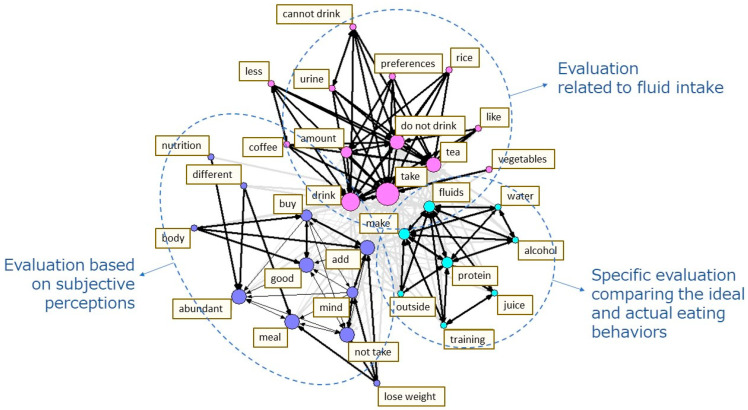
The co-occurrence network schema of the responses to the question “How do you evaluate your typical diet?”.

**Figure 3 nutrients-14-03720-f003:**
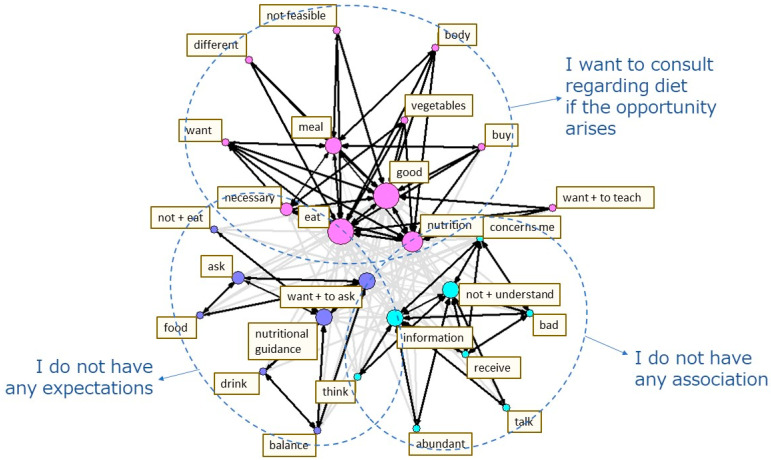
The co-occurrence network schema of the responses to the question “What role does the dietician play as your supporter?”.

**Figure 4 nutrients-14-03720-f004:**
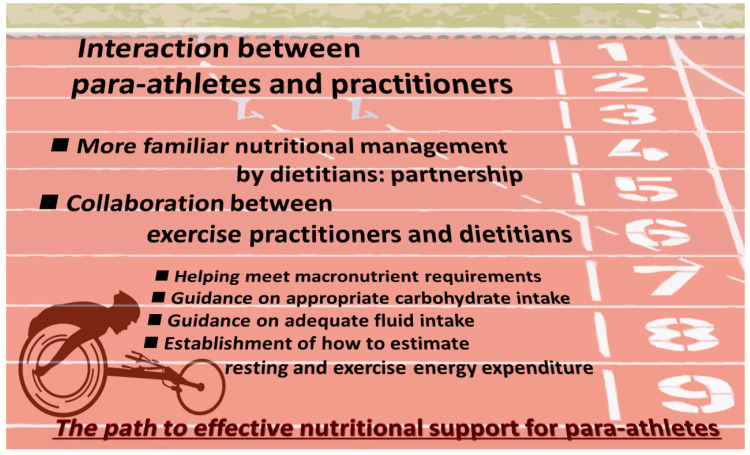
Path to effective nutrition support for para-athletes: understanding indispensable factors through interaction between para-athletes and practitioners.

**Table 1 nutrients-14-03720-t001:** Overview of study participants.

No.	Age (Years)	Gender	Event	Experience in Sport (Years)	Primary Disabilities	Ranking, etc.
1	44	F	Wheelchair tennis	20	Spinal cord injury	Japan Wheelchair Tennis Association Wheelchair tennis ranking 19th place (Women’s Singles, 2020)
2	37	M	Wheelchair softball (Class 3)	2	Spina bifida	Member of a team in the Japanese wheelchair softball league Experience in baseball for people with physical disabilities: 19 years Experience in wheelchair marathon: 4 years
3	45	M	Wheelchair softball (Class 3)	4	Spina bifida	Member of a team in the Japanese wheelchair softball league (Representative of the Japanese team in 2020) Experience in baseball for people with physical disabilities: 14 years Experience in wheelchair tennis: 4 years
4	32	M	Wheelchair softball (Class 2)	2	Spinal cord injury	Member of a team in the Japanese wheelchair softball league (Representative of the Japanese team in 2020) Experience in wheelchair basketball: 3 years
5	70	F	Wheelchair table tennis (Class 5)	26	Sacral tumor	Represented Japanese team in the Athens, Beijing, London, and Rio de Janeiro paralympic games
6	30	M	Wheelchair table tennis (Class 5)	9	Severe thermal trauma	2019 Japan Open Para Table Tennis Championship Individual gold medal, representative of the Japanese team in 2019 World ranking 35th place
7	30	F	Boccia (Physical Therapist)	---	Able-bodied	Boccia International Referee Japan Boccia Association Referee at Grade SA

**Table 2 nutrients-14-03720-t002:** Sentence composition of text data in the responses to each question.

Item	Value		
	(A)	(B)	(C)
	What is the ideal diet for improving competitive performance?	How do you evaluate your typical diet?	What role does the dietician play as your supporter?
Total number of responses	7	7	7
Mean character count per response	641	1225	1184
Total number of sentences	149	311	228
Mean character count per sentence	30	28	36
Total number of terms	871	1675	1596
Number by term category	427	7	673

**Table 3 nutrients-14-03720-t003:** Frequently occurring terms in the responses to each question (top 20 terms).

	(A)	(B)	(C)
What Is the Ideal Diet for Improving Competitive Performance?	How Do You Evaluate Your Typical Diet?	What Role Does the Dietician Play as Your Supporter?
Term	Number of Occurrences	Term	Number of Occurrences	Term	Number of Occurrences
Most frequent 	Consume	6	To consume	7	To consume	7
Not + consume	4	To drink	6	Good	7
Vegetables	4	Team	5	Nutrition	6
Good	4	Not + drink	5	Nutritional guidance	5
Calories	3	Meal	5	Information	5
Nutrition	3	Not + consume	5	Meals	5
Mind	3	Abundant	5	Not + understand	5
Like	3	Add	5	Want + to ask	5
Wheelchair	3	Good	5	Image	4
Meal	3	Protein	4	Necessary	4
Not + know	3	Mind	4	Ask	4
Not + good	3	Make	4	No + imagine	3
Side dishes	2	Water	4	Balance	3
Rice	2	Buy	4	Bad	3
Salad	2	Amount	4	Different	3
Balance	2	Alcohol	3	To drink	3
Different	2	Coffee	3	Am concerned	3
Eating out	2	Rice	3	Want + to teach	3
Don’t care	2	Juice	3	To think	3
Muscles	2	Training	3	To receive	3

**Table 4 nutrients-14-03720-t004:** Categorization of response details to the question “What is the ideal diet for improving competitive performance?”.

Categories	Subcategories	Examples of Responses That Contained Words inside the Clusters
(1) Strategically consuming specific nutrients	Importance of maintaining physical strength and concentration during matches	The **necessary nutrients differ** according to the sporting event I do not **need** large **muscles** for my sporting event Intake of sugars that increase agility and concentration rather than power is **necessary**
(2) Balanced intake	Vegetables should be eaten	It is best to eat **vegetables** I **add** a **vegetable** side dish when I **eat out**
	It is important to have a balanced diet	**Nutritional balance** is important Various **nutrients** should be **consumed** in a **balanced manner** Multiple food items **should be consumed**
(3) Meals that help in avoiding weight gain	Weight gain also impairs daily living	Regardless of whether body weight **can be measured on a scale**, reduce body fat so that the **wheelchair** can be pushed with a greater speed. This can be achieved by mainly having a **side dish**-based diet **Weight gain** is an issue as it affects the movement from **the wheelchair** to bed or car I **minimize the intake of rice** and other staple foods because a **wheelchair-based** lifestyle requires the intake of fewer **calories**
(4) Absence or lack of knowledge on the ideal diet for improving competitive performance	I do not know what the ideal diet is	There is an overwhelming amount of information when I search for “athletes’ **nutrition**,” so I **do not know** what I should **consume** in the end I **do not know** the **amount** of **food that should be taken** to maintain concentration during matches
	I have no ideals	I **eat** freely without **creating** rules; **I do not really mind** I just **pay attention** to the **amount** of food I eat to make sure that I **do not overeat**, but I generally like to eat what I want and the food that I like

Bolds represent the words which appeared in the co-occurrence network schema (Figure 1).

**Table 5 nutrients-14-03720-t005:** Categorization of response details to the question “How do you evaluate your typical diet?”.

Categories	Subcategories	Examples of Responses That Contained Words inside the Clusters
(1) Evaluation related to water intake	I am consuming adequate water	I believe that **water** is the most essential thing for the **body**; therefore, I **consume** 1.5 L of water everyday I consciously **drink water** to avoid **urinary** tract infections
	I drink less than the optimal amount of water	I limit water intake due to the fear of incontinence since I do not have the urge to **urinate** (due to my spinal injury) I do not feel thirsty, so I **cannot drink** water unless I consciously drink it. Thus, I believe my **water** intake is **less** than the ideal intake **Coffee** constitutes the majority of my daily **fluid intake**
(2) Specific evaluation comparing the ideal and actual eating behaviors	I am supplementing the necessary nutrients	I **consume** a lot of salads and **vegetable juices** I **consume protein** before and after the **training** or matches I want to **have** protein in my diet, so I ask my caregiver to **prepare** chicken side dishes
	I do not eat breakfast	Since I always had the habit of not eating in the morning, I **do not have** breakfast on most of the days
(3) Evaluation based on subjective perceptions	My physical condition is good	I believe in **maintaining** a proper **diet**, since my **physical** conditions are **good**. Since I make effort to **eat vegetables**, my physical conditions are **good** and I do not get tired easily
	I am able to minimize food intake	Our physical activity levels are **lower** than the able-bodied people, so I **pay attention** on my diet to avoid eating until I am full, and I am able to do so My mother (the in charge of cooking) serves too much on my plate, but I do not finish the entire food to reduce the **amount** I eat I try **not to eat** excessive **rice** in the evening I perform desk work during the day, so I try to reduce the **amount** of **meals** for breakfast and lunch
	I feel that my dietary intake is higher than the ideal amount	I want to **lose weight**, but I tend to eat **abundant amount** of foods in my **meals**

Bolds represent the words which appeared in the co-occurrence network schema (Figure 2).

**Table 6 nutrients-14-03720-t006:** Categorization of response details to the question “What role does the dietician play as your supporter?”.

Categories	Subcategories	Examples of Responses That Contained Words inside the Clusters
(1) I have the desire to consult regarding diet if the opportunity arises	I want to learn and expand my knowledge on nutrition	I **want** to learn how to use **vegetable** juices, health foods, and “food for specified health uses” I want to know the amount of energy **necessary** for wheelchair users I want someone to **teach** me the **necessary** amount of protein intake (the more you consume, the **better**?) I want someone to **teach me** the effective cooking methods that do not destroy the **nutrients** in **foods** I want someone to **teach me** the kind of **nutrients** that are **good** for **maintaining** focus during matches
	I want my diet to be managed	I want to gain knowledge on a menu of the **good** amount that I can **eat** (appropriate quantity) As I get older, I am having other health concerns related to my **body**, so I want my diet to be managed, including **nutritional balance** management and salt intake restrictions I want to know the **appropriate** way to allocate and decide the **nutrients** to be **consumed** between the three meals
(2) I do not have any association	I do not think I need nutritional guidance on a daily basis	The image that I have is **receiving nutritional guidance** only when I become ill or when I am hospitalized I obtain **nutrition**-related **information that concerns me** from the internet **Receiving** support from a personal dietician is financially **not feasible**
	I do not know how to get acquainted	**I do not know** where to contact when I want to **receive nutritional guidance**
(3) I do not have any expectations	I would not be able to put whatever I learn into practice	I do want to **hear** or take advice **about nutrition** from a dietician. This is because I do not think I can put it into practice since I cannot cook Even if I **received nutritional guidance**, that would be the end of it I cannot picture myself going out of my way to **receive nutritional guidance**

Bolds represent the words which appeared in the co-occurrence network schema (Figure 3).

## Data Availability

The data presented in this study are openly available in FigShare at http://doi.org/10.6084/m9.figshare.19745098.

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
