# Peer review of "The Role of Diets and Dietitians for Para-Athletes: A Pilot Study Based on Interviews"

_nutrients, 2022, doi:10.3390/nu14183720_

Round 1

Reviewer 1 Report

1. Reconsider and correct the title to ' The Role of Diets and Dietitians for Para-Athletes: A pilot study based on interview surveys' 2. Review your sentences and keep them , 3. Need more referencing throughout the article but especially in introduction (ie. Sports nutrition and contribution to para-athletes, what is happening in different countries) 4. New paragraph on line 71 will make the document to be more clear and better to read 5. Review Tables 3, 6, and 9 and combine them into one focusing on the goals of the survey 6. Over all , there are long paragraphs in the discussion which it will be useful to be revised 7. Are there any plans to elaborate the pilot study? State it 8. Elaborate on the concussions and state future study plans

Author Response

We appreciate Reviewer 1 for the kind and important comments. Our responses point by point are as below.

  1. Reconsider and correct the title to ' The Role of Diets and Dietitians for Para-Athletes: A pilot study based on interview surveys'

As Reviewer 1 pointed out, the title has been changed as follows.

“The Role of Diets and Dietitians for Para-Athletes: A Pilot Study Based on Interviews”

  1. Review your sentences and keep them.
  2. Over all , there are long paragraphs in the discussion which it will be useful to be revised

As Reviewer 1 pointed out, long paragraphs were hard to understand for readers. We split Paragraph 6 of the Discussion, which referred to the role of the dietitian/nutritionist, into three paragraphs. Similarly, we split Paragraph 7 into two paragraphs, and added a second paragraph about the potential development of this pilot study.

  1. Need more referencing throughout the article but especially in introduction (ie. Sports nutrition and contribution to para-athletes, what is happening in different countries)

Regarding the point, in some countries such as the United States and Australia, Dietetic Associations have issued statements and proposed guidelines regarding sports nutrition for para-athletes. However, such efforts are still limited, and even if guidelines have been proposed, there have not been enough attempts to apply them to para-athletes. Report concerning the contribution of sports nutrition to the competitive performance, conditioning and preventing injuries of para-athletes is scarce. Regarding the situation in Japan, there are very few sports dieticians who are specialized in parasports.

We mentioned the above in Introduction Paragraphs 3 and 4 (the revised manuscript) and added some references.

  1. New paragraph on line 71 will make the document to be more clear and better to read

According to Reviewer 1’s suggestion, we split the relevant paragraph into two paragraphs.

  1. Review Tables 3, 6, and 9 and combine them into one focusing on the goals of the survey

According to Reviewer 1’s suggestion, we merged Tables 3, 6, and 9 into a new Table 3 for the purpose of getting an overview of the terms and the characteristics of parts of speech contained in the responses of respondents.

  1. Are there any plans to elaborate the pilot study? State it
  2. Elaborate on the concussions and state future study plans

As Reviewer 1 pointed out, we added the description about the potential development of this pilot study to the 10th paragraph of revised manuscript.

Reviewer 2 Report

Title is confusing and grammatically incorrect. It should either be A pilot study based on an interview survey, or A pilot study based on interview surveys. I personally find the term 'interview survey' confusing and the word 'interviews' would be clearer.   

Abbreviations should only be used where they are very common. I would suggest not using the abbreviation PA and instead using the full word 'para-athletes"

The sentence in line 15-20 is very long and should be rephrased.

Line 49 - what is meant by 'practicing sports nutrition'?

Statements about the lack of nutritional advice for para-athletes are made in the paragraph starting on line 58. I am not convinced this is true for all countries and all sports. Please clarifiy if this is only about Japan, or about other specific countries too. 

I suggest that a new paragraph should start on line 71. The current paragraph is very long and covers several different aspects. Splitting into two paragraphs would help the reader.

Line 91 - 'thus' is a connecting word that suggests this should not be a new paragraph. It may be better to omit this word and start the paragraph without it.

Line 143 - please include the word language, to say 'language analysis software' as this then ties in better with the way it is described in the abstract.

It's not really clear why the physical therapist for the Boccia team is included in the interviews, this could be more clearly described. What is the rationale for including this one interviewee when all others are para-athletes?

Table 2. Means should be the same accuracy as the original measure - therefore the means should all the whole numbers, no decimals. Standard deviations should also be included.

Tables 3, 6, and 9 do not provide much insight into the nature of the responses. I would omit these and focus on some of the more useful data that is included. Examples of responses and a thematic analysis is much more useful. 

There are some very long paragraphs in the discussion that would be better split into smaller paragraphs and structured more carefully to make it easier for the reader to understand the interpretations being made.

The paper lends itself to practical recommendations for this group of athletes. I would expect to see a much stronger conclusion which gives these kinds of recommendations. I do not think that figure 4 is as useful as a good set of practical recommendations would be. The figure only repeats what is already presented in the paper.

It is positioned as a pilot study but there is no mention of the next steps for the full study. Will more data be collected in the same way? What changes will be made?  

With a stronger set of recommendations for practice this would be a useful and much improved paper.  There are plenty of interpretations in the discussion that would enable the authors to build this easily. 

Author Response

We appreciate Reviewer 2 for the kind and encouraging comments in detail. Our responses point by point are as below.

Title is confusing and grammatically incorrect. It should either be A pilot study based on an interview survey, or A pilot study based on interview surveys. I personally find the term 'interview survey' confusing and the word 'interviews' would be clearer.   

As Reviewer 2 pointed out, the title has been changed as follows.

“The Role of Diets and Dietitians for Para-Athletes: A Pilot Study Based on Interviews”

Abbreviations should only be used where they are very common. I would suggest not using the abbreviation PA and instead using the full word 'para-athletes"

We used ‘para-athletes’ instead of the abbreviation PA.

The sentence in line 15-20 is very long and should be rephrased.

We rephrased the sentense as suggested.

Line 49 - what is meant by 'practicing sports nutrition'?

As Reviewer 2 pointed out, the expression was unclear, therefore we revised “practicing sports nutrition” to “proper nutritional management based on sports nutrition”.

Statements about the lack of nutritional advice for para-athletes are made in the paragraph starting on line 58. I am not convinced this is true for all countries and all sports. Please clarifiy if this is only about Japan, or about other specific countries too. 

Regarding the point, in some countries such as the United States and Australia, Dietetic Associations have issued statements and proposed guidelines regarding sports nutrition for para-athletes. However, such efforts are still short, and even if guidelines have been proposed, there have not been enough attempts to apply them to para-athletes. Report concerning the contribution of sports nutrition to the competitive performance, conditioning and preventing injuries of para-athletes is scarce. Regarding the situation in Japan, there are very few sports dieticians who specialize in parasports.

I mentioned the above in Introduction Paragraphs 3 and 4 (the revised manuscript) and added some references.

I suggest that a new paragraph should start on line 71. The current paragraph is very long and covers several different aspects. Splitting into two paragraphs would help the reader.

According to Reviewer 2’s suggestion, we split the relevant paragraph into two paragraphs.

Line 91 - 'thus' is a connecting word that suggests this should not be a new paragraph. It may be better to omit this word and start the paragraph without it.

According to Reviewer 2’s suggestion, we deleted ‘thus’ from the beginning of the paragraph.

Line 143 - please include the word language, to say 'language analysis software' as this then ties in better with the way it is described in the abstract.

Thank you for the suggestion. We revised the words to 'language analysis software.'

It's not really clear why the physical therapist for the Boccia team is included in the interviews, this could be more clearly described. What is the rationale for including this one interviewee when all others are para-athletes?

Since the physical therapist works for a Boccia team, she is familiar with how players with problems with digestion and absorption due to cerebral palsy eat before and during games. We thought it would be useful for the study to interview her with such a perspective, therefore we decided to include her even though she is not a para-athlete. According to the suggestion by Reviewer 2, we added the above sentences to the revised manuscript.

Table 2. Means should be the same accuracy as the original measure - therefore the means should all the whole numbers, no decimals. Standard deviations should also be included.

As Reviewer 2 pointed out, the mean values in Table 2, 5, and 8 (the new combined Table 2) are expressed as whole numbers. Unfortunately, due to the characteristics of the software used in this study, the standard deviations are not calculated. Therefore, we cannot list the standard deviations in the Tables, we would like to replace this by sharing the full-text data of the interview responses in the online storage as described in L448 (the revised manuscript).  

Tables 3, 6, and 9 do not provide much insight into the nature of the responses. I would omit these and focus on some of the more useful data that is included. Examples of responses and a thematic analysis is much more useful. 

Since the examples of the original sentences contained in responses to each question can be referred in Tables 4, 5, and 6, We would like to leave the contents of Tables 3, 6, and 9 as they were in the original manuscript. However, we merged Tables 3, 6, and 9 into a new Table 3 for the purpose of getting an overview of the terms contained in the responses of respondents.

There are some very long paragraphs in the discussion that would be better split into smaller paragraphs and structured more carefully to make it easier for the reader to understand the interpretations being made.

It is positioned as a pilot study but there is no mention of the next steps for the full study. Will more data be collected in the same way? What changes will be made?  

As Reviewer 2 pointed out, long paragraphs were hard to understand for the reader. We split Paragraph 6 of the Discussion, which referred to the role of the dietitian/nutritionist, into three paragraphs. Similarly, we split Paragraph 7 into two paragraphs, and added a second paragraph about the potential development of this pilot study.

The paper lends itself to practical recommendations for this group of athletes. I would expect to see a much stronger conclusion which gives these kinds of recommendations. I do not think that figure 4 is as useful as a good set of practical recommendations would be. The figure only repeats what is already presented in the paper.

With a stronger set of recommendations for practice this would be a useful and much improved paper.  There are plenty of interpretations in the discussion that would enable the authors to build this easily. 

We really appreciate the important suggestion by Reviewer 2. We revised Figure 4 to include a set of practical recommendations instead of repeating nutritional challenges in para-athletes.